# LATTE: Learner-Adaptive Teacher-Forced Reflection for Advancing Deep Search

## Abstract

Deep search in LLMs hinges on efficiently acquiring external knowledge and up-to-date information to ground reasoning and generation. However, deep search agents often over-trust internal reasoning, terminate prematurely, and under-use external tools, resulting in brittle long-horizon performance. To address this, we introduce LATTE, a mixed-policy reinforcement learning framework that integrates teacher-forced, learner-adaptive reflection to provide oriented guidance that explicitly pushes the model to reflect, extend search rounds when evidence is insufficient, and increase the probability of beneficial tool calls. At each on-policy iteration, we seed reflective trajectories from the current policy's deep-search rollouts and inject teacher-forced critiques and corrections at decision points that govern whether to continue or stop the search and whether to defer to a tool or proceed with self-reasoning. By conditioning guidance on the learner's observed behavior and uncertainty, LATTE preserves on-policy updates while narrowing the gap between supervision and policy behavior, yielding an implicit curriculum focused on current failure modes (e.g., premature stopping, missed or delayed tool deferral, shallow exploration). Empirically, LATTE raises calibrated tool-use rates, lengthens effective search depth, and improves task success as well as training stability in advancing deep search optimization.

## 1 Introduction

Rapid progress in large language models Xi et al. (2025); Yao et al. (2023); Wang et al. (2024) has catalyzed the development of deep search agents Jin et al. (2025); Li et al. (2025d); Gao et al. (2025) that coordinate with external tools—search engines, code interpreters, and retrieval systems—to tackle long-horizon tasks. These tasks demand multi-step reasoning, selective information gathering, and robust decision-making under uncertainty. Despite progress, deep search agents frequently exhibit intertwined failure modes due to the inductive bias: they often over-trust their internal reasoning traces, terminate their search prematurely, and under-use external tools that could disambiguate uncertainty or verify hypotheses. The result is brittle performance when evidence is sparse, noisy, or distributed across multiple sources, especially under constrained tool budgets or time limits.

A central challenge is that existing training pipelines do not directly supervise the pivotal control decisions that govern search quality: whether to continue or stop the search, and whether to defer to a tool or proceed with self-reasoning. Standard supervised fine-tuning or RLVR typically focuses on per token likelihood or end-task accuracy rather than on the agent's meta cognitive process—its ability to recognize uncertainty, reflect on gaps, and adaptively escalate to tools. Moreover, supervision often occurs off-policy, e.g., post-hoc critiques Zhang et al. (2025), curated demonstrations Wan et al. (2025), which can misalign with the learner's actual on-policy behavior. This mismatch makes it difficult for agents to internalize reflection habits, learn calibrated stopping rules, and delineate the epistemic boundaries.

To address this issue, we introduce LATTE, a mixed-policy reinforcement learning framework that integrates teacher-forced, learner-adaptive reflection into on-policy deep search. Reflection (Shah et al., 2025; Yue et al., 2025; Gandhi et al., 2025) is a well-established augmentation for endowing LLMs with self-correction capabilities. In practice, a teacher model is employed to generate explicit post-thinking critiques that diagnose errors and synthesize corrective rules from responses of LLM. In LATTE, we incorporate reflection via teacher forcing within the on-policy optimization of deep

search agents, pushing them to self-reflect, back-trace the reasoning steps, and resume at failure decision points. Unlike existing agentic RL approaches that integrate reflection (Wu et al., 2025) by synthesizing reflective trajectories in an offline manner—thereby decoupling the teacher from the learner's state—, we propose a novel learner-adaptive reflection mechanism, in which the teacher's feedback is conditioned on the learner's current state. At each on-policy iteration, we seed reflective trajectories directly from the current policy's rollouts and inject oriented critiques as well as corrections at these decision points. Crucially, the learner-adaptive reflection, rather than "one-size-fit-all" supervision signals, allows LATTE to preserve the benefits of on-policy updates while narrowing the gap between supervision and policy behavior.

The key intuition is to couple exploration with structured, decision-centric feedback that is both timely (delivered at the moment of choice) and adaptive (conditioned on the learner's current failure modes). Rather than treating "reflection" as a generic prompt pattern or an offline annotation, LATTE operationalizes reflection as a controllable intervention in the agent's search loop. When the policy shows signs of premature stopping (e.g., high-variance beliefs, unsupported conclusions), the teacher-forced reflection pushes the search to continue. When the policy under-defers to tools despite uncertainty or conflicting evidence, the intervention increases the probability of a tool call.

LATTE is facilitated through a mixed-policy RL objective over a blend of on-policy and teacher-forced trajectories. We seed rollouts from the current policy's model, and inject reflections that consist of (i) step-by-step critiques (eg. cognitive shift, missing background knowledge, mistaken assumption), and (ii) corrective action plan corresponding to the critiques. This produces an implicit curriculum that naturally concentrates supervision on the learner's current weaknesses—premature stopping, missed tool calling, and shallow exploration—without drifting far from the on-policy distribution. By training on this mixture, the policy learns not only to produce better answers but also to internalize decision heuristics that generalize across tasks.

Empirically, LATTE raises tool-use rates, lengthens effective search depth, improves task success and training stability for advancing deep search RL optimization.

In summary, this work makes the following contributions:

- Proposes LATTE, a mixed-policy RL framework that integrates teacher-forced, learner-adaptive reflection at critical search control points.
- Preserves on-policy learning while narrowing the supervision–behavior gap through guidance conditioned on the learner's observed choices.
- Demonstrates improved performance on various benchmarks with self-reflection acquisition (e.g., calibrated tool-use, deepened effective search turns).
- Provides a general recipe to turn "reflection" from a generic prompt pattern into a trainable control mechanism within the search loop.

## 2 LATTE

LATTE is a on-policy RL training framework that couples explicit reflection to improve reasoning-centric language models. In Section 2.1, we introduce *Teacher-Forced Reflection*: trajectories are generated under teacher forcing strategy and enhanced through a learner-adaptive construction that conditions feedback on the model's current errors. In Section 2.2, we present *Mixed-Policy Optimization*: on-policy RL with GRPO is interleaved with reflection-augmented SFT. This combination of signal sources yields stable optimization and consistent gains across tasks.

### 2.1 TEACHER-FORCED REFLECTION

**Teacher-Forcing Strategy.** Deep search agents often exhibit undesirable inductive biases—prematurely answering with overconfidence, failing to call tools, or persisting with a wrong plan. To counter these behaviors during on-policy optimization, we introduce a teacher-forcing strategy that explicitly intervenes *within* rollouts rather than only reflecting post hoc. Let the policy be $\pi_\theta$, the input query be $q$, the ground-truth answer be $y^\star$, and the available tool set be $\mathcal{U}$. At step $t$, the model chooses an action

$$a_t \in \big\{ \text{answer}(y_t), \ \text{tool}(u_t, p_t) \big\}, \quad u_t \in \mathcal{U},$$

conditioned on the state $s_t$. $p_t \in \mathcal{P}(u_t)$ is denoted as the tool-call parameters required at step $t$. A *teacher* $T$ has privileged access to $y^\star$ and the step context; it monitors each action and only intervenes when necessary.

As illustrated in Fig. 1, the intervention rule is minimal and targeted:

- If $a_t$ is a tool call, no intervention is applied; the rollout proceeds normally.

- If $a_t$ is an answer action and $y_t \neq y^\star$, the model has made a cognitive error (e.g., missing evidence or faulty derivation). At this *moment*, the teacher injects a structured reflection $r_t$ into the context and enforces the *next* decision $a_{t+1}$ to be a tool call. Concretely, we append $r_t$ and apply a decoding constraint that masks out direct-answer actions, ensuring $a_{t+1} = \text{tool}(\cdot)$.

- If $a_t$ is an answer and $y_t = y^\star$, the episode terminates with success.

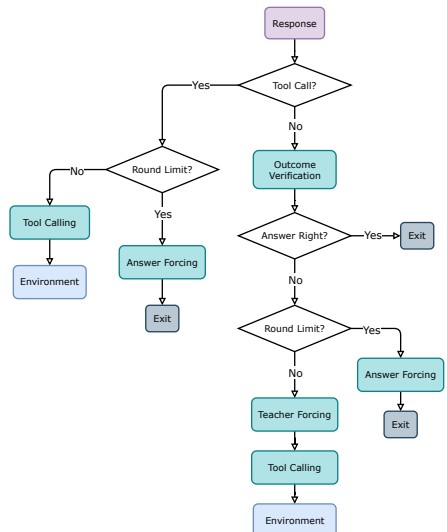

Figure 1: Flowchart of teacher-forcing strategy in on-policy RL.

This online, step-level teacher forcing encodes an *act→reflect* rhythm directly into trajectories: the model learns to recognize when its internal knowledge is insufficient and to switch to external tools before committing to an answer. By supervising the *decision boundary*—"answer now" versus "gather evidence first"—rather than micromanaging solution content, the policy internalizes a disciplined pattern of tool-first reasoning in uncertain states.

Implementation-wise, we realize the constraint via a control token or logit bias that forbids answer-type actions at $t+1$ and optionally prioritizes a teacher-suggested tool $u^\dagger \in \mathcal{U}$. The rollout thus contains tuples $(s_t, a_t, o_t, r_t)$ where $r_t$ is empty unless an incorrect answer triggered reflection, and $o_t$ denoted as the observation returned after executing tool call $u_t$. These trajectories are then used by our mixed-policy optimizer (Sec. 2.2).

**Learner-Adaptive Reflection.** Prior works Wu et al. (2025) typically generate reflections *offline* and *post hoc*, often summarizing errors over a full trajectory. Such reflections are temporally misaligned with the policy's actual decision points and may overfit errors that no longer occur as the policy improves. In contrast, LATTE produces reflection *online at every round* from the *current* policy state and its on-policy rollouts. This yields timely and targeted guidance that adapts to $\pi_\theta$ as it changes, maximizing alignment with on-policy updates and with the policy's realized behavior.

Concretely, at step $t$ with state $s_t$, we form a structured reflection

$$r_t = \big(c_t,\ \kappa_t\big),$$

with two components:

1. **Critique on rollout** $c_t$: a diagnosis of the model's *current* cognitive errors extracted from on-policy evidence. Rather than relying on offline generated responses, we sample a response $\tau_t$ from on-policy rollouts, thus $c_t$ is produced as:

$$c_t = T_c(\tau_t \mid s_t)$$

and the teacher $T$ analyzes the model's ongoing "thinking" and actions to identify concrete failure modes (e.g., premature answering, missing evidence, invalid derivation, tool misuse or misparameterization, hallucinated claims). The teacher then supplies a succinct corrective idea that addresses these errors.

2. **Action Calibration** $\kappa_t$: a decision-level correction that calibrates the imminent action choice,

$$c_t = T_\kappa(c_t \mid s_t, \tau_t)$$

This correction is minimal but actionable, steering the next decision without over-constraining the solution path.

Crucially, both $c_t$ and $\kappa_t$ are conditioned on the *current* policy via on-policy evidence. As $\theta$ evolves, the distribution of reflections $r_t$ co-evolves because it is a function of the learner's present behavior. This adaptivity minimizes covariate shift between the states seen during optimization and those used to generate guidance, thereby improving stabilization of on-policy updates.

Integration with the teacher-forcing rule (Fig. 1) is straightforward. Whenever the policy attempts an incorrect direct answer ($y_t \neq y^\star$), the teacher injects $r_t = (c_t, \kappa_t)$ and enforces that the following action is a tool call. In practice, $\kappa_t$ can populate a suggested tool $(u^\dagger, p^\dagger)$ that the decoding constraint prioritizes, while the critique $c_t$ provides just enough rationale to correct the identified failure mode. The policy then proceeds as

$$a_{t+1} \sim \pi_\theta\big(\,\cdot\,|\,s_t,\ r_t\big).$$

Although the mechanism is defined at every step, the intervention remains minimal: if the model is correct or already seeking evidence, $r_t$ can be empty and no constraint is applied.

This learner-adaptive design delivers three benefits:

- **Timely and targeted.** Reflections are produced exactly at the decision points where the model erred or hesitated, using on-policy rollouts rather than stale, offline traces.
- **Policy-state aligned.** Because $r_t$ depends on $\pi_\theta$, guidance evolves with the learner, maintaining alignment with the model's *current* inductive biases and error modes.
- **Act–reflect habit formation.** By repeatedly conditioning decisions on concise $r_t$, the policy internalizes a *do→reflect→revise* rhythm—learning to detect insufficiency early, seek evidence, and correct itself quickly.

In summary, learner-adaptive reflection transforms reflection from an offline, trajectory-level commentary into an online, state- and policy-aware scaffold. This keeps supervision focused on the *decision boundary*—when to answer versus when to gather evidence—while preserving exploration in how the answer is ultimately derived.

## 2.2 MIXED-POLICY OPTIMIZATION

The reflective trajectories introduced in the previous provide tool-call-centric feedback aligned with the agent's current policy, yet their integration into training requires a reinforcement learning algorithm that can (i) preserve the on-policy nature of updates, (ii) handle a heterogeneous mixture of rollouts (policy-driven and teacher-forced), and (iii) maintain stability under long-horizon credit assignment. To this end, we adopt a mixed-policy optimization strategy—GRPO with SFT—which allows optimization over a mixture of learner rollouts and teacher-forced reflections without incurring large distribution shifts.

For each mini-batch, we optimize a gated mixture of on-policy and supervised objectives:

$$\mathcal{L}(\theta) \;=\; \alpha\,\mathcal{L}_{\mathrm{RL}}(\theta) \;+\; \beta\,\mathcal{L}_{\mathrm{SFT}}(\theta)$$

The gate enforces on-policy updates only when the batch contains informative (non-constant) returns, while allows SFT optimization vice versa, i.e.,

$$\alpha = \begin{cases} 1 & \text{if all rollouts fail or succeed} \\ 0 & \text{otherwise} \end{cases} \quad, \quad \beta = \begin{cases} 0 & \text{if all rollouts fail or succeed} \\ 1 & \text{otherwise.} \end{cases}$$

Specifically, we adopt standard GRPO (Shao et al., 2024) for $\mathcal{L}_{\mathrm{RL}}$,

$$\mathcal{L}_{\mathrm{RL}}(\theta) = \frac{1}{Z} \sum_{i=1}^{N_{\mathrm{on}}} \sum_{k=1}^{|\tau_i|} \mathrm{CLIP}\big( \frac{\pi_\theta(\tau_{i,k}|q, \tau_{i,<k})}{\pi_{\theta_{\mathrm{old}}}(\tau_{i,k}|q, \tau_{i,<k})}, A_i, \epsilon\big),$$

and extend GRPO objective for reflection supervision through calibrating gradient estimates following Yan et al. (2025),

$$\mathcal{L}_{\mathrm{SFT}}(\theta) = \frac{1}{Z} \sum_{j=1}^{N_{\mathrm{ref}}} \sum_{k=1}^{|\tau_j|} \mathrm{CLIP}\big( \frac{\pi_\theta(\tau_{j,k}|q, \tau_{j,<k})}{\pi_\phi(\tau_{j,k}|q, \tau_{j,<k})}, A_j, \epsilon\big)$$

Table 1: **Results on HLE and GPQA (higher is better).** We compare LATTE against closed-source references (top), open 32B approaches and search-augmented systems (middle), and our 7B variants (bottom). LATTE-7B attains the strongest GPQA among reported open models (72.1) and improves over its non–teacher-forced variant on both benchmarks (+4.1 HLE, +3.2 GPQA). Dashes indicate results not reported.

| MODEL | HLE | GPQA |
|---|---|---|
| OpenAI-o3 | 20.2 | – |
| Claude-4-Sonnet | **20.3** | – |
| Qwen2.5-32B-Instruct | 5.4 | 48.0 |
| QwQ-32B | 9.6 | 65.6 |
| Search-o1 Li et al. (2025c) | 10.8 | 63.6 |
| ASearcher-Web-QwQ Gao et al. (2025) | 12.5 | – |
| WebThinker-32B Li et al. (2025d) | 15.8 | – |
| LATTE-7B w/o Teacher-Forced Reflection | 7.8 | 68.9 |
| LATTE-7B | 11.9 | **72.1** |

where $Z = \sum_{i=1}^{N_{\text{on}}} |\tau_i| + \sum_{j=1}^{N_{\text{ref}}} |\tau_j|$ is the normalization factor,

# 3 EXPERIMENT

## 3.1 EXPERIMENT SETUP

**Benchmarks.** We conduct a evaluation of our web agent on a science-oriented benchmarks, focusing on HLE Phan et al. (2025), a frontier benchmark with extremely challenging STEM problems, and GPQA-Diamond Rein et al. (2024), which targets graduate-level science reasoning.

**Implementation Details.** We use the Qwen2.5-7B model (Qwen et al., 2025) to conduct RL training, resulting in the LATTE-7B model. For reinforcement learning, we adopt GRPO (Guo et al., 2025) as the RL algorithm, we train on approximately 3K samples using the GRPO algorithm, where each group consists of 16 rollouts with a batch size of 128 and a learning rate of 1e-6. We set the turn limit as 32 for 7B. Our training is based on ASearcher (Sheng et al., 2025). For search tools, we follow previous work (Li et al., 2025a; Tao et al., 2025), our agent scaffold integrates both search and browse capabilities. The search tool issues one queries at each turn and retrieves the top-5 Google results with titles, URLs, and snippets, while the browse tool takes a URL and a query, retrieves the page content via Jina (Jina.ai, 2025), and leverages GPT-oss (Comanici et al., 2025) to answer based on the retrieved content.

**Baselines.** In our evaluation, we consider three recent search-augmented reasoning agents, namely WebThinker Li et al. (2025d), Search-o1 Li et al. (2025c), and ASearcher Gao et al. (2025), which represent state-of-the-art paradigms that integrate external information retrieval into the reasoning process. To provide a fair comparison with models that do not rely on search tools, we further prompt Qwen-2.5-32B-Instruct and QwQ-32B to directly generate answers without invoking any external resources, thereby isolating the intrinsic reasoning capabilities of large-scale LLMs. In addition, we include closed-source models (e.g., OpenAI-o3 and Claude-4-Sonnet) as strong baselines.

**Evaluation.** We adopt LLM-as-Judge (LasJ) as the primary evaluation metric, where a strong LLM (GPT-oss) is prompted to assess the correctness of model outputs under task-specific instructions. Our results are reported with Avg@4.

## 3.2 OVERALL RESULTS

**Web-based Search and Browsing on QA Benchmarks.** Table 1 summarizes results on HLE and GPQA. Despite being a 7B model, LATTE-7B achieves the strongest GPQA among reported open systems, reaching 72.1—surpassing QwQ-32B (65.6; +6.5) and the search-augmented Search-o1 (63.6; +8.5), and far exceeding Qwen2.5-32B-Instruct (48.0; +24.1). On HLE, LATTE-7B attains 11.9, improving over open 32B baselines and some search-augmented systems (Qwen2.5-32B: 5.4;

Table 2: **Ablation study for Different forcing methods.** We compare three methods: (i) *S1-Style Forcing*: prefixes a brief self-correction trigger (e.g., "Oh wait, ...") ; (ii) *Prompt Hint*: adds a lightweight answer hint together with the query; and (iii) *Teacher-Forcing*: the proposed strategy in LATTE. Best scores are highlighted in **bold**.

| Forcing Method | GPQA | | HLE | |
|---|---|---|---|---|
| | **Avg@4** | **Pass@4** | **Avg@4** | **Pass@4** |
| S1-Style Forcing | 54.6 | 83.3 | 6.4 | 18.5 |
| Pormpt Hint | 56.8 | 83.3 | 6.0 | 16.6 |
| Teacher-Forcing | **66.7** | **86.4** | **8.2** | **20.0** |

Table 3: **Ablation study for Different Reflection Strategies.** We compare three strategies in the source selection of the initial response in reflection trajectory construction: (1) *Adaptive-Self* generates reflections with the current policy (evolves with updates); (2) *Frozen-Self* uses a frozen snapshot of the same policy at initialization; (3) *Frozen-Other* uses frozen snapshots of external policies (Qwen3-8B/32B). We report Avg@4 and Pass@4 on GPQA and HLE; higher is better. Best results are highlighted in bold

| Reflection Strategy | GPQA | | HLE | |
|---|---|---|---|---|
| | **Avg@4** | **Pass@4** | **Avg@4** | **Pass@4** |
| Adaptive-Self (Qwen2.5-7B) | 66.7 | **86.4** | **8.2** | **20.0** |
| Frozen-Self (Qwen2.5-7B) | **66.8** | 83.8 | 8.0 | 19.0 |
| Frozen-Other (Qwen3-8B) | 65.2 | 84.9 | 7.5 | 17.8 |
| Frozen-Other (Qwen3-32B) | 59.2 | 85.4 | 7.5 | 19.6 |

+6.5, QwQ-32B: 9.6; +2.3, Search-o1: 10.8; +1.1), while trailing larger search-augmented 32B models such as WebThinker-32B (15.8) and ASearcher-Web-QwQ (12.5). Comparison with baseline model indicate that teacher-forced reflection is a key contributor: compared to the non–teacher-forced variant, LATTE-7B improves by +4.1 HLE ($7.8 \rightarrow 11.9$) and +3.2 GPQA ($68.9 \rightarrow 72.1$).

## 3.3 ABLATION STUDY

**Different Forcing Methods.** We compare three forcing strategies in Table 2: (i) *S1-Style Forcing*, which prepends a brief self-correction trigger; (ii) *Prompt Hint*, which appends a lightweight answer hint to the query; and (iii) *Teacher-Forcing*, our proposed strategy in LATTE. Across both GPQA and HLE, Teacher-Forcing yields the best performance on all metrics. On GPQA, it improves Avg@4 to 66.8, surpassing S1-Style and Prompt Hint by +12.2 and +10.0 points, respectively, while marginally increasing Pass@4 to 86.4 (vs. 83.3 for both baselines). On HLE, Teacher-Forcing also attains the highest Avg@4 (8.2; +1.8 over S1-Style and +2.2 over Prompt Hint) and the best Pass@4 (20.0; +1.5 and +3.4, respectively). We hypothesize that S1-Style Forcing mainly encourages superficial self-revision signals without reliably steering the reasoning trajectory, and Prompt Hint can introduce bias or premature commitment to hinted patterns—both of which may limit consistency gains. In contrast, Teacher-Forcing explicitly constrains intermediate rollouts to align with high-quality trajectories, stabilizing multi-step reasoning and improving average correctness. These results validate the choice of Teacher-Forcing in LATTE as the most effective forcing mechanism among those considered.

**Different Reflection Strategies.** As shown in Table 3, our method (*Adaptive-Self*) consistently delivers the strongest success under multi-try evaluation while maintaining virtually the same average quality as the best frozen alternative. On GPQA/HLE, Adaptive-Self achieves the top Pass@4 of $86.4/20.0$, while matching the best Avg@4 within only $0.1 - 0.2$ points (66.7 vs. 66.8 on GPQA). Compared to the strongest self baseline (Frozen-Self), this translates to $+2.6/ + 1.0$ gains in Pass@4 on GPQA/HLE at a negligible Avg@4 cost of $0.1$. In the common setting where Pass@k is the primary objective, Adaptive-Self thus attains a strictly more favorable operating point.

Why does adaptivity help? Co-evolving the reflector with the learner keeps reflections *on-policy*—calibrated to the current decoding distribution and stylistic conventions—thereby reducing mismatch and yielding more targeted, actionable critiques. This improves the chance that at least one of the $k = 4$ attempts succeeds, without sacrificing average quality. Evidence comes from the

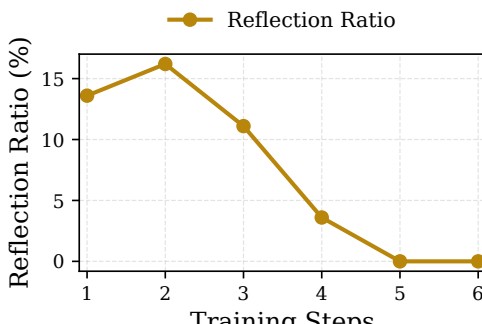 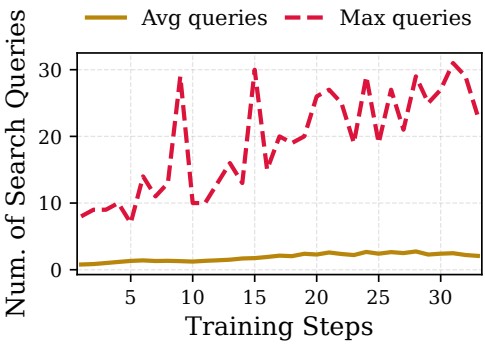

(a) Reflection Token Ratio over Training Steps.  (b) Number of Search Queries over Training Steps.

Figure 2: Training dynamics of reflection and tool use. (a) Reflection Token Ratio: the proportion of tokens devoted to reflective critique vs. solution content at each training step. Under LATTE with learner-adaptive teacher-forced reflection, the ratio adapts over time, indicating on-policy calibration that steers multi-step reasoning without inducing superficial self-revision. (b) Number of Tool Calls: the average count of tool calls (search queries) per step. The controlled evolution of query counts shows that improvements stem from better-targeted and more stable rollouts.

*Frozen-Other* variants: using external frozen policies (Qwen3-8B/32B) degrades overall performance. Qwen3-8B reduces both Avg@4 and Pass@4 on GPQA and HLE, while the stronger Qwen3-32B recovers Pass@4 (85.4/19.6) but severely harms GPQA Avg@4 (59.2; $-7.5$ vs. Adaptive-Self), indicating a reflection-style and calibration mismatch that our on-policy approach avoids.

## 4 ANALYSIS

### 4.1 QUANTITATIVE ANALYSIS

**Analysis of Reflection Ratio.** To quantify how interventions evolve as the policy improves, we measure the teacher-forcing trigger rate, defined as the fraction of reflection trajectory tokens evolved in optimization over all tokens in a step. Across the optimization steps as shown in Fig. 2a we observe several trends emerge:

- *Self-dampening schedule.* Teacher forcing is most active early and rapidly recedes as the policy adapts, dropping latter with the update going. This indicates that interventions are aligned with learning progress: as $\pi_\theta$ internalizes the guidance, it needs less external correction.

- *On-policy distribution shift.* The small uptick from early rounds suggests transient exploration effects from on-policy updates (GRPO + reflection-augmented SFT), after which the policy quickly stabilizes with far fewer erroneous early answers and better tool-first behaviors under uncertainty.

**Analysis of Tool Call.** We analyze how tool-call frequency evolves over optimization as shown in Fig. 2b. Empirically, the average number of tool calls consistently increases along with policy updates. This trend matches the intended effect of Teacher-Forced Reflection: when the model proposes an incorrect answer, the teacher injects structured reflection and masks direct-answer logits, increasing the probability of a subsequent tool call action. As training proceeds, the policy learns a calibrated deferral strategy—calling tools more readily under uncertainty or knowledge gaps. The slight pullback after the peak likely reflects improved base competence: on easier instances the model resolves queries with fewer calls, while on harder ones it still escalates to tools when needed. The maximum number of tool calls per episode also expands over training, indicating deeper exploratory chains become accessible under teacher-forced reflection.

**Question** In Olya's black box, there are 5 apples and 7 pears, and in the white box, there are 12 pears. In one move, Olya can blindly take a fruit from any box and either eat it or move it to the other box. Olya wants the contents of the boxes to be the same. What is the minimum number of moves Olya can guarantee to achieve this? The boxes are closed, and the fruits cannot be identified by touch.

**Response** Okay, let's try to figure out this problem. So, Olya has two boxes: a black one with 5 apples and 7 pears, and a white one with 12 pears. The goal is to make the contents of both boxes the same with the minimum number of moves. Each move allows her to take a fruit from any box, either eat it or move it to the other box. The boxes are closed, so she can't tell which fruit she's taking.

First, I need to understand what "contents of the boxes to be the same" means. Does it mean the number of fruits in each box must be equal, or that the composition (number of apples and pears) must be the same? The problem says "the contents of the boxes to be the same, " which probably refers to the number of each type of fruit. But maybe it's about the total number of fruits? Wait, the original counts are black box has 5+7=12 fruits, white has 12 pears. So total fruits are 12 + 12 = 24. If they need to have the same contents, maybe each box should have 12 fruits, but the problem says "the contents of the boxes to be the same, " which might mean that each box has the same number of apples and pears. But the white box only has pears. So maybe the question is about making the number of each type of fruit equal in both boxes?

**Reflection - Critique** Oh, Wait. In my initial response, I misunderstood the problem's requirement for "the contents of the boxes to be the same. " I assumed it meant equalizing the number of apples and pears in both boxes, but I overlooked the fact that the total number of apples (5) is odd, making it impossible to split equally without removing some fruits. This led me to ignore the possibility of eating fruits as part of the solution. Additionally, I didn't properly model the worst-case scenario for blind moves, where Olya might repeatedly pick the wrong type of fruit (e.g., pears instead of apples), increasing the required number of moves. I also failed to calculate the exact number of moves needed to adjust both apple and pear counts while accounting for the necessity of eating certain fruits to achieve parity. This reflects a gap in considering both combinatorial constraints and the strategic use of eating as a tool to balance totals.

**Reflection - Calibration** Next, I need to clarify the exact problem requirements by verifying if "same contents" refers to equal fruit counts per type or total fruits. Then, I must model the worst-case scenario for blind moves, considering the necessity of eating fruits to resolve parity issues. This involves calculating the minimal moves required to ensure the desired state regardless of fruit selection. I will use the web search tool to find similar combinatorial problems or mathematical strategies for this scenario.

Next step action: <search> minimum moves to equalize boxes with apples and pears blind draws </search>

Figure 3: Case study illustrating how reflection mitigates overconfidence. While the initial roll-out over-committed to a symmetry-based solution and overlooked hidden constraints, reflection introduced both uncertainty confirmation and algebraic consistency checks, enabling the agent to reformulate the problem into degree equalities and verify them systematically across committees.

## 4.2 CASE STUDY

To analyze the effect of our method, we present a focused case study on LATTE, illustrating how the approach operates in practice and the kinds of improvements it enables.

**Overconfidence Correction.** A demonstrative case study (Fig. 3) shows a correction on overconfidence failure modes observed in the initial rollout: the initial response prematurely locked onto symmetry and produced an incorrect total, without validating constraints implied by the "exactly 10 common acquaintances and 10 common non-acquaintances" condition. Reflection inserts uncertainty confirmation and algebraic consistency checks, which force the agent to translate the English constraints into equalities over committee-wise degrees, and to verify them across all permutations of the three committees.

## 5 RELATED WORK

**Deep Search.** Recent work has sought to enhance LLM-based agents by equipping them with external tool use. Early prompt-based systems such as Search-o1 (Li et al., 2025c), MindSearch (Chen et al., 2024) and ReAgent (Zhao et al., 2025) enabled rapid prototyping but were limited by model capacity and lack of feedback adaptation. To improve generalization, some studies synthesized retrieval–reasoning trajectories for SFT (Asai et al., 2023; Yu et al., 2024), while others explored reinforcement learning (RL) on multi-hop QA benchmarks like HotpotQA and 2WikiMultihop, showing gains in tool usage and reasoning (Jin et al., 2025; Song et al., 2025; Chen et al., 2025; Zheng et al., 2025; Li et al., 2025b). More recently, researchers have begun to focus on more challenging tasks, by fine-tuning sophisticated prompt-based agents powered by Large Reasoning

Models through offline RL (Li et al., 2025d), SFT on simulated trajectories with real-world web data (Li et al., 2025a; Sun et al., 2025), and constructing challenging QAs for RL training (Tao et al., 2025; Liu et al., 2025). In addition, several studies, such as ASearcher (Gao et al., 2025), demonstrate that extending the number of search tool calls can further improve agent performance. While these approaches mainly rely on reinforcement learning with autonomous rollouts to encourage increased tool usage, our work takes a different direction. We propose a reflective trajectory synthesis pipeline that alleviates the common issues of over-trusting internal reasoning and under-utilizing external tools. By integrating reflection into trajectory generation, our method not only promotes more beneficial tool calls and improves sample efficiency, but also enables the model to acquire self-reflection and error-recovery capabilities.

**Reflection in RL.** A growing body of research highlights the role of reflection as an essential ingredient in reinforcement learning with LLMs. Early work has primarily focused on outcome-based reward optimization, which implicitly encourages models to revisit their reasoning chains and adjust subsequent actions, leading to emergent self-correction behaviors (Guo et al., 2025). Furthermore, recent studies underscore the critical importance of the inherent capabilities and behaviors present in the base models before task-specific fine-tuning or reinforcement learning begins. Research indicates that foundational abilities for verification and reflection are not merely helpful but often prerequisites for successful online learning and significantly influence the ultimate performance ceiling achievable through RL (Shah et al., 2025; Yue et al., 2025; Gandhi et al., 2025). Beyond textual reasoning, reflection-based RL has recently been extended to multimodal domains, particularly in visual mathematical reasoning. For instance, VL-Rethinker (Wang et al., 2025) introduces structured critique–revision loops to refine problem-solving steps, while Critique-GRPO Zhang et al. (2025) and SRPO incorporate explicit reflection modules into the reinforcement optimization process, showing that reflective signals can substantially improve the robustness of reasoning with complex visual inputs. These advances suggest that reflection not only improves sample efficiency and error recovery in text-based agents but also offers a promising pathway to strengthen multimodal LLMs where reasoning must integrate symbolic and perceptual information.

## 6 LIMITATION

While our mixed-policy RL framework with teacher-forced, learner-adaptive reflection is designed to sharpen decision-centric control in deep search agents, its effectiveness ultimately hinges on the quality and calibration of teacher critiques at the moments where guidance is injected. Noisy or biased feedback can distort learned stopping and deferral thresholds, pushing the learner toward maladaptive behaviors—over-extending search, over-deferring to tools, or stopping prematurely. Mismatches between the teacher's uncertainty calibration or domain coverage and the deployment environment can further imprint miscalibrated heuristics, degrading reliability under distribution shift. One remedy is to employ more capable teacher models, but this also introduces greater computational and operational overhead. We leave this trade-off to future work.

## 7 CONCLUSION

In this paper, we introduce LATTE, a mixed-policy reinforcement learning framework that integrates teacher-forced, learner-adaptive reflection at critical decision points. LATTE aligns supervision with on-policy behavior throughing seed reflective trajectories from the current policy's rollouts and inject step-wise critiques and corrective action plans *at the moment of choice*, thereby coupling exploration with timely, decision-centric feedback while preserving the benefits of on-policy updates. Empirically, LATTE raises calibrated tool-use, lengthens effective search depth, improves task success, enhances sample efficiency, and stabilizes training compared to deep search baselines that lack learner-adaptive, teacher-forced reflection. These results indicate that supervising the *meta-cognitive* control of search—rather than only end outputs—enables agents to better recognize uncertainty, back-trace errors, and adaptively escalate to external tools.

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
