# OpenReview forum: "LATTE: Learner-Adaptive Teacher-Forced Reflection for Advancing Deep Search"
_ICLR.cc/2026/Conference — ICLR 2026 Conference Withdrawn Submission_

### Official Review · Reviewer_UYqT · 2025-10-25

**Soundness:** 3
**Presentation:** 2
**Contribution:** 2
**Rating:** 4
**Confidence:** 4

**Summary:**

This paper proposes LATTE, a mixed-policy reinforcement learning framework that integrates teacher-forced, learner-adaptive reflection into deep search agents. The key innovation is injecting structured reflection at critical decision points (whether to continue/stop search, whether to use tools) during on-policy rollouts. Unlike prior work that generates reflections offline, LATTE conditions teacher feedback on the learner's current policy state, producing "learner-adaptive" reflections. The method combines GRPO for on-policy optimization with reflection-augmented trajectories. Experiments on HLE and GPQA show improvements over baselines, with LATTE-7B achieving 72.1 on GPQA and 11.9 on HLE.

**Strengths:**

1.	Clear conceptual contribution: The distinction between "learner-adaptive" (on-policy) and "frozen" (off-policy) reflection is well-articulated, with good intuition for why on-policy reflection matters.
2.	Practical framework: The teacher-forcing strategy with explicit intervention rules (Figure 1) is simple and implementable, making the work reproducible.
3.	Solid ablation studies: Tables 2 and 3 provide meaningful comparisons of forcing methods and reflection strategies, supporting design choices.

**Weaknesses:**

1.	Limited technical novelty: The core contribution is essentially combining existing techniques (GRPO + teacher forcing + reflection) with an "on-policy" twist. The algorithmic innovation is incremental—the main difference from prior work is when and how reflections are conditioned, not fundamentally new methods.
2.	Narrow experimental validation: (1) Only 2 benchmarks (HLE, GPQA), both science-focused QA; (2) No evaluation on open-ended research tasks
3.	Computational costs not reported: Training time, inference latency, cost of teacher model calls, API costs for web searches—all missing. This is critical for assessing practicality
4.	Limited analysis of when/why it helps: (1) Case study (Figure 3) shows one success example but no failure analysis; (2) No breakdown by question difficulty or type; (3) No analysis of what types of errors the reflection catches vs. misses
5.	Unclear generalization: What about other tools or information sources?

**Questions:**

1.	Teacher model specification: What model serves as the teacher T? How is it trained/selected? What are the computational costs?
2.	On-policy vs off-policy: Can you provide direct evidence that on-policy reflection is better because of distribution alignment, not just because of using a different/better teacher?
3.	Reflection frequency: How often does teacher forcing actually trigger? Figure 2a shows it decreases—does this mean the method becomes less important later in training?
4.	Comparison with Meta-Researcher: Meta-Researcher achieves 73.2 GPQA (Table 1 of their paper), higher than LATTE's 72.1. How does your method compare on the same setup?
5.	Ablation on RL: What happens with just SFT on teacher-forced trajectories, without RL? Is RL necessary?
6.	Tool call quality: Does increasing tool calls always help? Could the method lead to unnecessary tool overuse?
7.	Frozen reflections with better teacher: If you use a stronger frozen teacher (e.g., GPT-4), would that close the gap with adaptive reflection?

---

### Official Review · Reviewer_aGcw · 2025-10-26

**Soundness:** 3
**Presentation:** 4
**Contribution:** 2
**Rating:** 6
**Confidence:** 1

**Summary:**

This paper introduces LATTE, a mixed-policy reinforcement learning framework that integrates teacher-forced, learner-adaptive reflection at critical decision points during on-policy rollouts. Rather than relying solely on end-to-end rewards or offline supervision, LATTE dynamically seeds reflective trajectories from the agent’s own behavior and injects step-wise critiques and corrective action plans precisely when decisions are made. This approach tightly couples exploration with timely, decision-centric feedback, while preserving the stability and credit assignment advantages of on-policy learning.

Empirically, LATTE demonstrates significant improvements over deep search baselines lacking such adaptive reflection: it enhances calibrated tool use, extends effective search depth, boosts task success rates, improves sample efficiency, and stabilizes training. These gains highlight a key insight—supervising the meta-cognitive control of reasoning (e.g., when to reflect, when to backtrack, when to invoke tools) is more effective than supervising only final outcomes. By enabling agents to better recognize uncertainty, trace back errors, and adaptively escalate to external resources, LATTE advances the development of more self-aware and strategically capable language agents in RLVR settings.

**Strengths:**

(1) Well-written (2) Detailed experiment (3) The problems related to training efficiency that have been solved are distinctive and seem valuable to the industrial sector

**Weaknesses:**

N/A

**Questions:**

I do not know this specialized research field very well. I will adjust my score and optimize my review document based on the evaluations of other expert reviewers and my performance during the rebuttal period.

---

### Official Review · Reviewer_mrbg · 2025-10-31

**Soundness:** 2
**Presentation:** 3
**Contribution:** 2
**Rating:** 4
**Confidence:** 3

**Summary:**

1. This paper presents LATTE, a mixed-policy RL framework for LLM deep search, addressing issues like over-trusting internal reasoning via teacher-forced, learner-adaptive reflection.
2. It injects targeted critiques and corrective actions at decision points during on-policy training, using a mixed objective (GRPO-based RL + SFT) to align supervision with the learner’s behavior.
3. On HLE and GPQA benchmarks, LATTE-7B outperforms open-source 32B models and search-augmented baselines, boosting tool use, search depth and task success.

**Strengths:**

1. Self-reflection is a key ability for large reasoning models
2. The experiments validate the effectiveness of the proposed method

**Weaknesses:**

1. The proposed on-policy Teacher-Forcing Strategy seems to be a decoding strategy that utilizes the outcome reward. It may improves the trajectory better, but it is not reasonable to run RL on these trajectories, since it is off-policy (the behavior policy is changed due to the decoding strategy).
2. There are some related works that tune LLMs to learn self-reflection, which should be considered for comparison.
3. The evaluation should include more challanging benchmarks for robust evaluation.

**Questions:**

See Weakness

---

### Note · Authors · 2025-12-02

I have read and agree with the venue's withdrawal policy on behalf of myself and my co-authors.